

# Quantifying the current and future likelihood of the 2022 extreme wildfires weather conditions in France with anthropogenic climate change

Shengling Zhu[1], Renaud Barbero[1], François Pimont[2], and Benjamin Renard[1]

[1]INRAE, Aix Marseille University, RECOVER, Aix-En-Provence, France
[2]INRAE, URFM, Avignon, France

**Correspondence:** Shengling Zhu (shengling.zhu@inrae.fr)

**Abstract.** In 2022, southwestern France experienced an exceptional wildfire season, recording a burned area 14 times higher than the 2006–2023 average. Here, we assess the rarity of the fire weather conditions observed in 2022 and how anthropogenic climate change (ACC) has already altered and will continue to alter the probability of fire weather conditions associated with the three largest wildfires (Landiras-1: 12,552 ha; Landiras-2: 7,124 ha; La Teste-de-Buch: 5,709 ha). Drawing from the

daily Fire Weather Index (FWI) computed from two reanalysis datasets (1959-2023) and a nationwide wildfire record dataset (2006–2023), we first sought to quantify the rarity of those conditions across a range of spatial (fire location versus regional) and temporal (fire duration versus monthly) scales. Our results show that the extremeness of FWI conditions generally increases with the spatiotemporal resolution, with the associated return periods increasing from 6 to 34 years, from 22 to 89 years, and from 6 to 101 years when moving from the coarsest to the finest spatiotemporal scale for the Landiras-1, Landiras-2, and

La Teste-de-Buch wildfires, respectively. Using climate simulations from the Coupled Model Intercomparison Project Phase 6 (CMIP6), we examined how ACC has modified the probability of such fire weather conditions between 1950 and 2100. We found that by 2022, ACC at least doubled the likelihood of those FWI conditions, and will make them, by the end of the century (under the Shared Socioeconomic Pathway 2-4.5, (SSP2-4.5)) at least 10–100 times more probable, depending on the models. Our study underlines the growing influence of ACC in the risk of extreme wildfires in France across a range of scales.

## 1 Introduction

The past decade has witnessed a number of unprecedented extreme wildfires across parts of the world (e.g. Australia in 2019–2020, Canada in 2023, or California in 2025), causing widespread impacts on societies, ecological environments, and human life. In 2022, southwestern Europe also faced an extreme fire season due to a persistent anticyclonic anomaly (Faranda et al., 2023) causing widespread soil moisture deficit (Bevacqua et al., 2024), and record burned area in some regions (Ro-

drigues et al., 2023) including parts of southwestern (SW) France. Across France, more than 55,000 hectares of forests and other natural vegetation were burned (IGN and MASA, 2025) – an area 6 to 7 times larger than the average over the preceding decade and more than 14 times larger than the average in SW France (Fig. 1b). This extensive burned area resulted in substantial biomass losses in Atlantic pine forests (Vallet et al., 2023) and was largely driven by a small number of wildfires. In



particular, three events alone accounted for more than 45 % of the total annual burned area in France in 2022 and over 80 %

of that in SW France. On 12 July 2022, two wildfires started simultaneously within the Gironde department: La Teste-de-Buch wildfire burned approximately 6,000 hectares over 12 days, while the second one in Landiras burned over 12,552 hectares over 14 days, due to frequent wind shifts causing spread in multiple directions. After this wildfire (hereafter Landiras-1) was brought under control, it reignited one month later, on 9 August 2022 (hereafter Landiras-2), and spread over six days, driven by northerly winds. When combined, the Landiras-1+2 wildfire burned over 19 776 ha, which makes it the largest wildfire in

France since the 1940s.

Those wildfires in southwestern France provided a glimpse of future projections across the region, featuring a spatial expansion of the potential fire niche with climate change towards western and northern latitudes (Fargeon et al., 2020), a fire niche historically limited to the southeastern Mediterranean region. The 2022 fire season was indeed concomitant with a broader context of global and regional climate warming. Copernicus data indicate that July 2022 was among the three warmest Julys

recorded globally, exceeding the 1991–2020 average by about 0.38 °C. In France, Meteo-France recorded an average annual temperature of  14.5 °C, approximately 2.9 °C higher than the 1959–2000 baseline. This warmer atmosphere and elevated atmospheric aridity has contributed to reduce fuel moisture content, thereby increasing landscape flammability.

Beside climate–fire studies, attribution analysis is essential to better understand how global warming is currently altering the likelihood of extreme events and associated impacts (Perkins-Kirkpatrick et al., 2024). Quantifying the likelihood of such

impacts may enhance awareness and encourage adaptation efforts. Attribution studies employ both observational and simulated climate datasets to quantify the extent to which human emissions alter the probability of a given extreme weather event. So far, most of those studies have typically focused on meteorological events such as heatwaves (Perkins-Kirkpatrick and Lewis, 2020; Vautard et al., 2020), droughts (Chiang et al., 2021; Hari et al., 2020), or extreme rainfall (Tradowsky et al., 2023; Wang et al., 2023). However, fire weather conditions (combining multiple meteorological variables) have received less attention,

although a number of efforts have been made in North America (Abatzoglou and Williams, 2016; Williams et al., 2019; Brown et al., 2023), Canada (Kirchmeier-Young et al., 2019a), Australia (van Oldenborgh et al., 2021), or in France (Barbero et al., 2020; Lanet et al., 2024). Recently, Lanet et al. (2024) conducted an attribution study of the 2022 fire season in SW France using multiple standardized climate indices. Their findings suggest that climate change doubled the likelihood of the climate conditions observed during the month of July. However, the analysis was performed over relatively broad spatial (the entire SW

region of France) and temporal (the whole month of July) scales, while attribution scores have been shown to be sensitive to the selection of spatial and temporal scales (Angélil et al., 2018; Kirchmeier-Young et al., 2019b; Leach et al., 2020). Finally, the attribution scores were limited to the year 2022, with no projections on how those conditions might change in the future. Such projections may help plan adaptation strategies.

Building on prior attribution studies, we use here a complementary multi-scalar framework to (i) provide a broader context of

the spatiotemporal variability of fire weather conditions across France over the whole observational period from 1959 to 2023, (ii) quantify fire weather anomalies conducive to the 2022 wildfires relative to the 1959–2023 baseline, (iii) estimate return periods (RPs) of fire weather conditions associated with the three largest wildfires, considering a combination of different and complementary spatial and temporal scales, and (iv) estimate the extent to which those fire weather conditions have become





more or less likely due to human-induced climate change in 2022 and how this contribution is likely to further increase in the
future.

## 2   Data and methods

### 2.1   Wildfire data

We used the BDIFF (Base de Données sur les Incendies de Forêts en France, IGN and MASA (2025)) dataset, a forest-fire
record for France (2006–2023), providing date, location, and burned area (BA; ha). Despite some consistency issues in low
fire activity regions (Pimont et al., 2023a), BDIFF has been shown as reliable for estimating regional total BA, including the
SW France, and has been used in previous studies (Pimont et al., 2021, 2023b). We selected wildfires ≥ 1 ha occurring during
the warm season (May to September, see Fig. 1a) following previous studies (Pimont et al., 2021). We also classified wildfires
into three size classes (small: 1–10 ha; medium: 10–100 ha; large: > 100 ha) as shown in Fig. 1a. Note that large wildfires
were mostly concentrated in southern France, in particular along the Mediterranean coast and SW France (Fig. 1a). In 2022,
national BA reached approximately 55 000 ha, with the SW region accounting for over half of this total (Fig. 1b) due to three
major wildfires. Table 1 provides the name, starting and ending dates as well as the extent of each of those wildfires.

| Fire name | Burning period | Burned area (ha) | Contribution (%) |
|---|---|---|---|
| Landiras-1 | 12–25 July 2022 | 12 552 | 40 |
| Landiras-2 | 9–14 August 2022 | 7 124 | 22 |
| La Teste-de-Buch | 12–23 July 2022 | 5 709 | 19 |

**Table 1.** The three largest wildfires in southwest France in 2022. The last column indicates their respective contribution to total burned area
in southwest France in 2022.





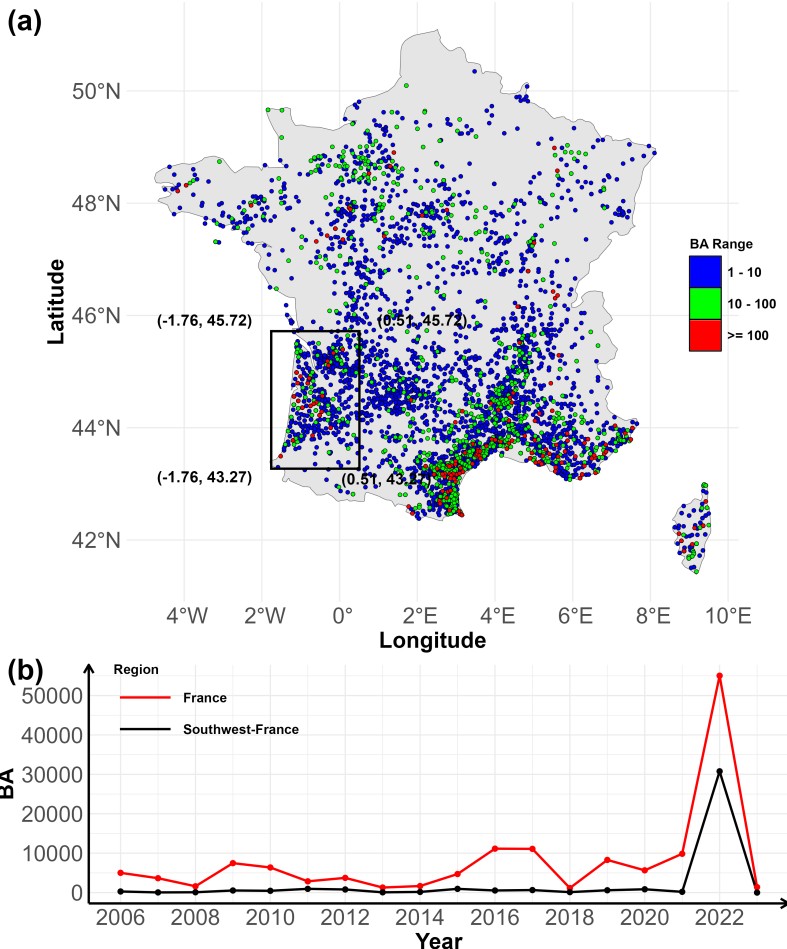

**Figure 1.** (a) Wildfires $\geq 1$ ha recorded in the BDIFF database from 2006 to 2023. Color denotes burned area (BA) classes and the black box delineates the southwestern France region ($\sim 4.9 \times 10^4$ km$^2$). (b) Total burned area during the warm fire season (May–September) for France (red line) and the southwestern region (black line).

## 2.2 Fire weather observations

Fire weather conditions were estimated with the Fire Weather Index (FWI), a composite index based on four daily meteo-
rological observations – maximum air temperature, relative humidity, wind speed, and precipitation. Originally developed in
Canada by Wagner (1987) for northern boreal forest conditions, the FWI has been used across various countries and climatic
regions to track wildfire activity, including in Europe (Giannaros et al., 2021; Hetzer et al., 2024) and in France (Barbero
et al., 2020; Pimont et al., 2023a, b). Daily meteorological variables used to calculate the FWI were obtained from SAFRAN
(Système d'Analyse Fournissant des Renseignements Atmosphériques à la Neige), a French reanalysis product available at a





daily resolution on an 8 km grid from 1959 to 2023 (Vidal et al., 2010). The analyses were repeated using ERA5 reanalysis at a slightly coarser resolution of 25 km. The results were similar for both datasets (see Fig. S1–S3 in the Supplement). The FWI was calculated using the CFFDRS package in R (Wang et al., 2017).

## 2.3    Spatiotemporal variability of fire weather

In climate sciences, empirical orthogonal function (EOF) is often employed to examine large spatiotemporal datasets and
identify the main modes of climate variability. In this study, an EOF was applied to a matrix with seasonal FWI values (averaged from May to September, corresponding to the traditional wildfire season) for each grid cell across France, structured as an $n_{\text{year}} \times n_{\text{grid}}$ matrix, where $n_{\text{year}}$ is the number of years and $n_{\text{grid}}$ the number of grid cells. After data normalization and eigenvalue decomposition, the matrix was decomposed into a few dominant spatial modes (EOFs), together with their corresponding time-varying coefficients, known as principal components (PCs). Each PC constitutes a time series that illustrates the interannual
variability of its corresponding EOF (von Storch and Zwiers, 1999).

## 2.4    Fire weather conducive to wildfires in SW France

To quantify the relationship between local FWI conditions and fire events, we extracted for each wildfire the daily FWI time series in the grid cell co-located with the wildfire over a window extending from 90 days before to 90 days after the wildfire start. To quantify how unusual were those conditions, we removed the seasonality by computing deviations from the local
mean seasonal cycle and expressed them as percentage anomalies. Finally, we stratified the sample by fire size and averaged FWI within predefined fire extent classes (small, medium, large) to relate the amplitude of FWI anomalies to fire size. Finally, analyses were performed on all years (2006–2023) and then on the year 2022 only.

## 2.5    Fire weather simulations

The Coupled Model Intercomparison Project Phase 6 (CMIP6) provides a comprehensive and standardized ensemble of multi
model climate simulations, enabling improved understanding of climate change driven by natural internal variability and ex ternal radiative forcings under various past, present, and future scenarios (Eyring et al., 2016). Here, we used simulations from the Detection and Attribution Model Intercomparison Project (DAMIP) and the Scenario Model Intercomparison Project (ScenarioMIP) (Gillett et al., 2016; O'Neill et al., 2016), a sub-project of CMIP6. DAMIP provides simulations for historical peri ods (up to 2014, with extensions to 2020 in some cases) under anthropogenic and natural forcings scenarios (Gillett et al.,
2016). ScenarioMIP provides climate projections informed by future emissions and land-use scenarios, primarily driven by Shared Socioeconomic Pathways (SSPs) (O'Neill et al., 2016; Riahi et al., 2017). For historical simulations, we used the "historical" and the DAMIP "hist-nat" experiments. The "historical" experiment covers 1850–2014 and includes all observed external forcings – greenhouse gases, aerosols, solar variability, and volcanic eruptions. In contrast, the "hist-nat" experiment includes only natural external forcings (total solar irradiance and volcanic stratospheric aerosol injections) over 1850–2020. For
future climate projections, we used the "ssp245", representing a medium mitigation scenario and "ssp245-nat" experiments.



Similar to their historical counterparts, the "ssp245" experiment includes both anthropogenic and natural forcings, whereas the "ssp245-nat" experiment includes only natural forcings. Not all CMIP6 models provide all the meteorological outputs needed to compute the FWI (see Sect. 2.2). Here, we used the following models: IPSL-CM6A-LR, CanESM5, MIROC6, and NorESM2-LM.

## 2.6 Probability of exceedance of 2022 extreme wildfires across spatial and temporal scales

We quantified the expected return periods of fire weather conditions associated with the top-3 largest fires in 2022 to assess the rarity of these conditions. This estimation requires three steps: 1) characterizing fire weather conditions associated with each wildfire across spatial and temporal scales; 2) fitting an appropriate statistical distribution; 3) calculating the exceedance probability (or return period) of the fire weather conditions defined in step 1 thanks to the distribution fitted in step 2.

The choice of temporal and spatial scales is the most critical step due to their impact on attribution scores (Leach et al., 2020; Kirchmeier-Young et al., 2019b). Moreover, refining or broadening scales may provide different and complementary insights for wildfire managers. In the temporal dimension, we may either opt for a 30-day window centered on the fire occurrence as in Lanet et al. (2024) or focus on fire duration from ignition to suppression, a period more representative of the burning conditions. In the spatial dimension, we may either select a single grid cell (8-km with SAFRAN or 25-km with ERA5) co-located with the wildfire location or consider a broader regional bounding box as done in Lanet et al. (2024) to improve the signal-to-noise ratio. Here, in order to analyze the sensitivity of results to these assumptions, we used the four possible combinations of these different resolutions. For each wildfire, we derived the daily FWI time series corresponding to both spatial resolutions over the full period (1 January 1959–31 December 2023). We then applied a moving average (MA) to each time series, using both a 30-day window and a $D$-day window – where $D$ equals the fire duration (Landiras-1: 14 days; Landiras-2: 6 days; La Teste-de-Buch: 12 days). The annual maxima of these MA time series were then extracted to fit the historical distribution using the generalized extreme value (GEV) theory. Finally, based on the fitted distributions, we calculated the exceedance probabilities and corresponding return periods of the observed FWI during the top-3 largest wildfires in 2022 according to each spatiotemporal scale. Note that in this case, the FWI level associated with each wildfire does not necessarily match with the annual maxima (see for instance Fig. 5).

## 2.7 The contribution of anthropogenic climate change

To quantify the impact of anthropogenic climate change (ACC), we employed a commonly used approach to calculate the exceedance probability of each wildfire-related FWI ($p_{\mathrm{OBS}}$) (Barbero et al., 2020), following the procedure applied in the previous section to the 1959–2023 SAFRAN observations. We then compared the exceedance probabilities under two scenarios: (i) the ALL scenario, including all anthropogenic and natural forcings (hereafter $p_{\mathrm{ALL}}$), and (ii) the NAT scenario, which includes only natural forcings (hereafter $p_{\mathrm{NAT}}$). For the GEV distribution fitted to the ALL-scenario simulated annual maxima of the MA-FWI time series, we inverted its cumulative distribution function (CDF), $F_{\mathrm{ALL}}(x)$, to find the FWI level in the $p_{\mathrm{ALL}}$ scenario such that $1 - F_{\mathrm{ALL}}(\mathrm{FWI}_{\mathrm{ALL}}) = p_{\mathrm{OBS}}$. We then applied this same threshold $\mathrm{FWI}_{\mathrm{ALL}}$ to the GEV distribu-



tion fitted to the NAT-scenario simulated annual maxima of the MA-FWI time series – using its CDF $F_{\text{NAT}}(x)$ – to compute $p_{\text{NAT}} = 1 - F_{\text{NAT}}(\text{FWI}_{\text{ALL}})$.

As the FWI is assumed to be non-stationary with global warming, we used here a nonstationary GEV model where the location and scale parameters may vary with year according to a GAMLSS framework (Generalized Additive Models for Location, Scale and Shape), to capture smooth nonlinear relationships (Stasinopoulos and Rigby, 2007; Rigby et al., 2019). The covariate effects on the location and log-scale parameters are represented using penalized cubic regression splines in the `mgcv` package, with at most five basis functions for each smooth term, and smoothing parameters selected by restricted

maximum likelihood (REML; Youngman, 2022). As opposed to what has been done previously in Sect. 2.6 where we sought to estimate the probability of exceedance over the full observational period available without any assumptions, we did use here a non-stationary GEV to make the return periods explicitly time-dependent. Based on the GAMLSS fits, we estimated for each year the GEV distribution of annual maxima FWI under the ALL and NAT scenarios and computed the ratio $\frac{p_{\text{ALL}}}{p_{\text{NAT}}}$, commonly referred to as the risk ratio (RR). This metric has been widely used in event-attribution studies to quantify how many times

as likely an extreme event is to occur under the ALL scenario compared to the NAT scenario (Philip et al., 2020; Paciorek et al., 2018). Additionally, we employed the fraction of attributable risk ($\text{FAR} = 1 - 1/\text{RR}$), which reflects, when positive, the proportion of risk attributable to ACC (Philip et al., 2020; Lloyd and Oreskes, 2018; Bellprat et al., 2019). Finally, attribution scores (RR and FAR) from individual models were aggregated using a multi-model median across models.

    To quantify the sampling uncertainty surrounding RR and FAR, a parametric bootstrap approach was implemented as fol-

lows:

1. Generate new samples of ALL and NAT scenarios from the estimated non-stationary GEVs.

2. Re-estimate the non-stationary GEVs based on these new samples and compute the RR and the FAR.

3. Repeat steps 1-2 100 times to derive parametric confidence intervals.

    The full computation workflow—from the CMIP6 annual maxima MA-FWI to the RR/FAR calculation—is summarized in

Fig. 2.





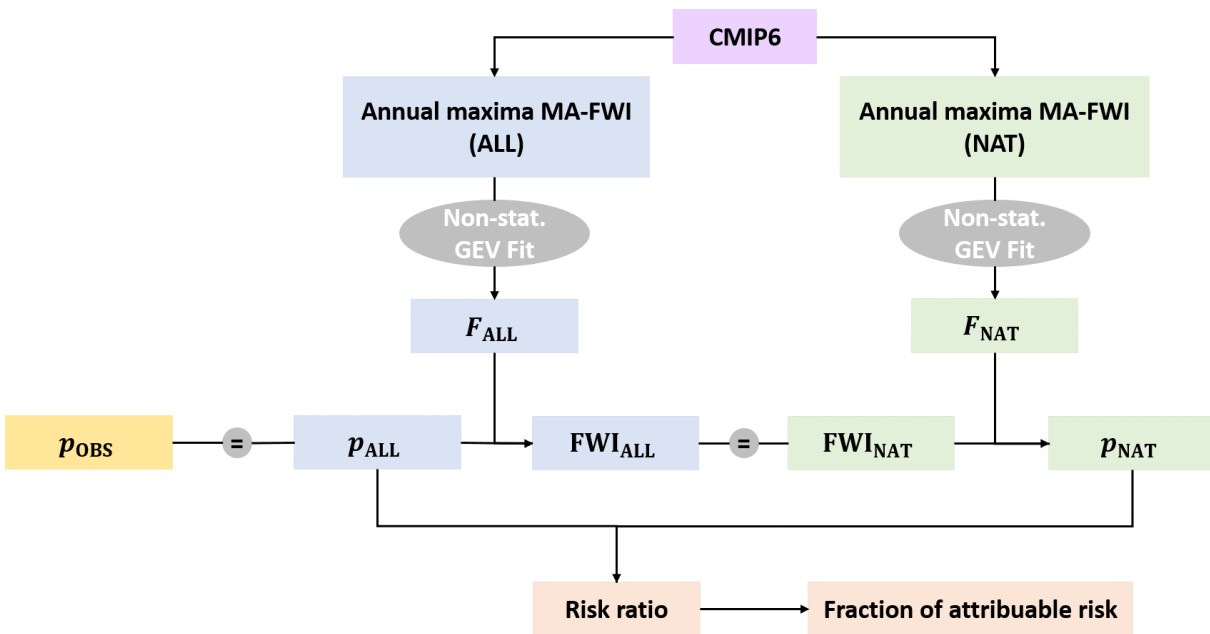

**Figure 2.** Schematic workflow used to estimate exceedance probabilities (denoted by $p$) under ALL and NAT forcings from CMIP6 annual maxima MA-FWI: a non-stationary GEV fit provides the cumulative distribution functions (CDFs) $F_{\mathrm{ALL}}(x)$ and $F_{\mathrm{NAT}}(x)$ (where $F$ denotes the fitted GEV CDF); $\mathrm{FWI}_{\mathrm{ALL}}$ is obtained by inverting $F_{\mathrm{ALL}}$ such that $1 - F_{\mathrm{ALL}}(\mathrm{FWI}_{\mathrm{ALL}}) = p_{\mathrm{OBS}}$, and $p_{\mathrm{NAT}}$ is then computed as $1 - F_{\mathrm{NAT}}(\mathrm{FWI}_{\mathrm{ALL}})$.

## 3    Results

Figure 3 illustrates the first two modes of May–September FWI over the observational 1959–2023 period with the spatial distributions of EOF loadings (left panel) and their corresponding principal components (right panel). Together, the first two
modes explain about 75 % of the total variance (62 % and 13 % for EOF-1 and EOF-2, respectively). EOF-1 exhibits positive loadings throughout France, albeit with smaller coefficients along the Mediterranean (Fig. 3a). This mode presents a strong interannual variability with an upward underlying trend, featuring an increasing frequency of higher FWI years in recent decades with global warming (Fig. 3b). Note that the highest amplitude is seen in 2022, followed by 1976, a notoriously warm and dry year in France. By contrast, after removing the influence of PC-1, EOF-2 shows a slightly unbalanced north–south
dipole, with a near-zero band straddling central France. Note that this mode can be viewed as a mostly Mediterranean mode given the amplitude of the EOFs loadings along the Mediterranean (Fig. 3c). In other words, when positive FWI anomalies occur preferentially in the south, negative anomalies are seen in the north and vice versa. Like PC-1, PC-2 presents a long-term trend (Fig. 3d), reflecting an increasing occurrence of years with higher FWI in southern France and lower FWI in northern France. Note that the following modes were not analyzed due to their little variance explained and their lack of consistency
across space.





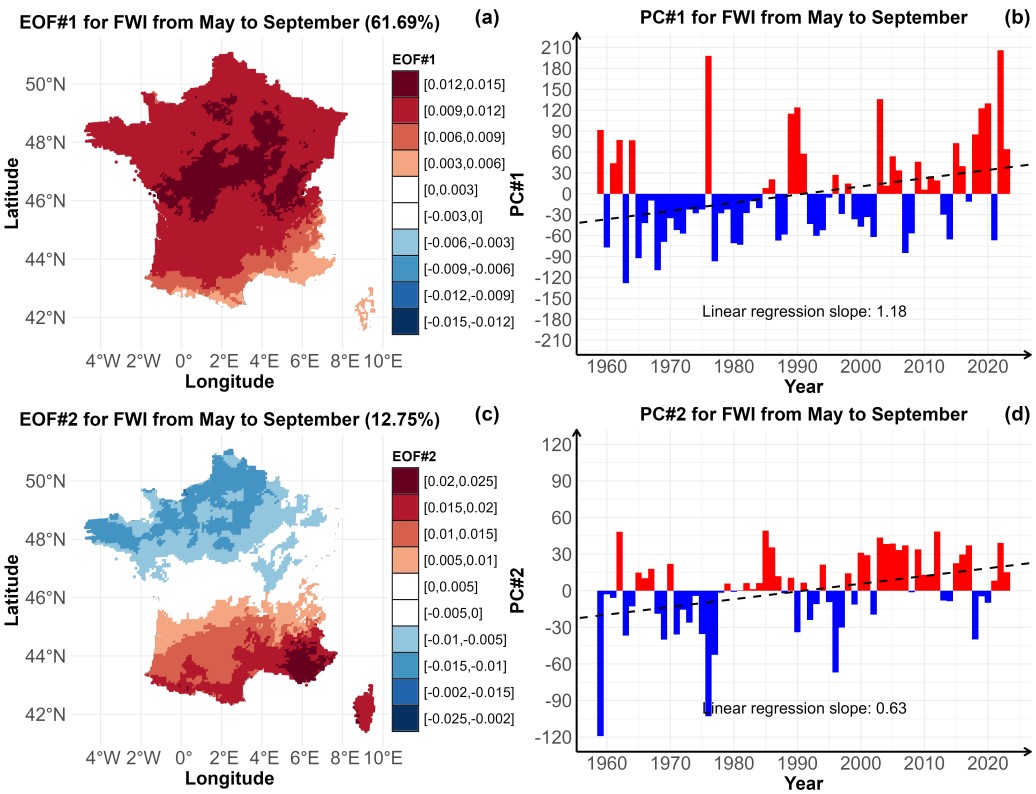

**Figure 3.** Leading two modes of mean May–September FWI over France from 1959 to 2023. First EOF (a) (variance explained $62\%$) with its corresponding PC time series (b). Second EOF (variance explained $13\%$) (c) with its corresponding PC time series (d).

We then restricted our attention to local FWI conditions associated with actual wildfires across SW France. Figure 4a shows that FWI increases until the wildfire day and decreases in the following days, with an amplitude proportional to fire size (i.e. higher FWI for larger fires). Figure 4c shows positive anomalies 3 months before fires (note that $100\%$ indicates FWI was twice larger than what we would expect from the average local conditions), reaching 71 %, 106 %, and 155 % for small, medium and large fires, respectively. A similar signal was observed in 2022 (Fig. 4b,d), but FWI anomalies were that time stronger during the previous months and were 119 %, 137 %, and 180 % higher than mean conditions the starting days of small, medium and large wildfires, respectively (Fig. 4d).



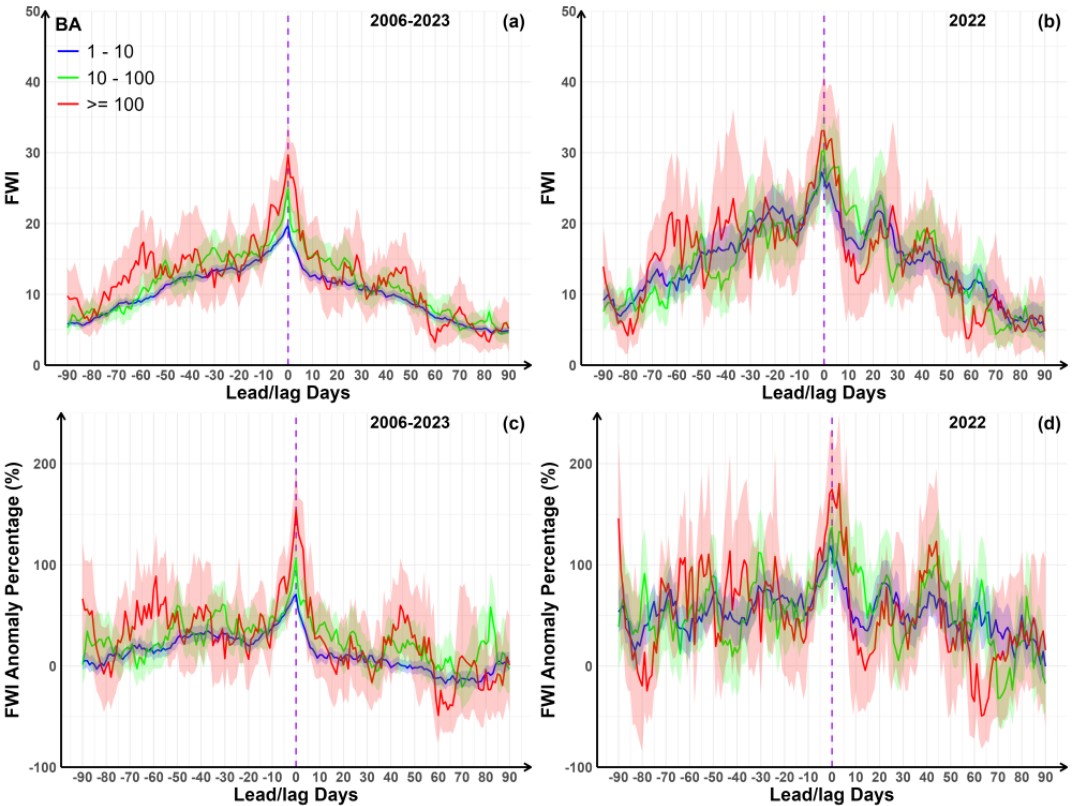

**Figure 4.** Lead–lag time series of FWI (a,b) and percent anomalies (c,d) relative to wildfire dates for three fire size classes over 2006–2023 (a,c) and 2022 only (b,d) in SW France. Anomalies were computed relative to the long-term (1959–2023) mean local seasonal cycle. Blue, green, and red curves denote BA = 1–10 ha, BA = 10–100 ha, and BA ≥ 100 ha, respectively. Shaded bands indicate 95 % bootstrap confidence intervals. The $x$-axis shows lead/lag days from $-90$ to $+90$ relative to the wildfire starting day (day 0; purple dashed line).

We then examined the return periods (RPs) of the top-3 largest wildfires of 2022. Figure 5 (left panels) indicates that annual maxima of the MA FWI are consistently highest when computed at the finest spatiotemporal resolution (i.e., the fire-duration window at the 64 km$^2$ SAFRAN grid cell fire level) and decrease when either the temporal window is lengthened to 30 days or when FWI conditions are averaged over the SW France region ($\sim 4.9 \times 10^4$ km$^2$). This pattern holds for all three wildfires (Fig. 5, left panels). In every case, the absolute maximum occurs in 2022, underscoring the exceptional FWI conditions during

that year. The estimated RP of the observed annual maxima FWI associated with the three wildfires (Fig. 5, right panels) span the following ranges across the four spatiotemporal scale: Landiras-1: 6–34 years; Landiras-2: 17–89 years; La Teste-de-Buch: 6–101 years, illustrating how sensitive the RPs are to the chosen scales. Overall, the rarity of those conditions also increases with the resolution, with RPs increasing from ≈6 to ≈34 years, from ≈22 to ≈38 years, and from ≈6 to ≈101 years when moving from the coarser to the finest spatiotemporal scale for Landiras-1 (Fig. 5b), Landiras-2 (Fig. 5d), and La Teste-de-Buch

(Fig. 5f) wildfires, respectively.







**Figure 5.** Annual maxima of moving-averaged FWI at multiple spatiotemporal scales (left) and return-levels (right) associated with Landiras-1 (a-b), Landiras-2 (c-d), and La Teste-de-Buch (e-f). Return levels on the right (logarithmic x-axis from 1 to $10^3$ years) were estimated by fitting a GEV distribution to the annual-maximum moving-averaged FWI. Shaded envelopes indicate 80 % parametric-bootstrap confidence intervals. Dashed lines indicate the estimated return periods of the FWI level observed for each wildfire.





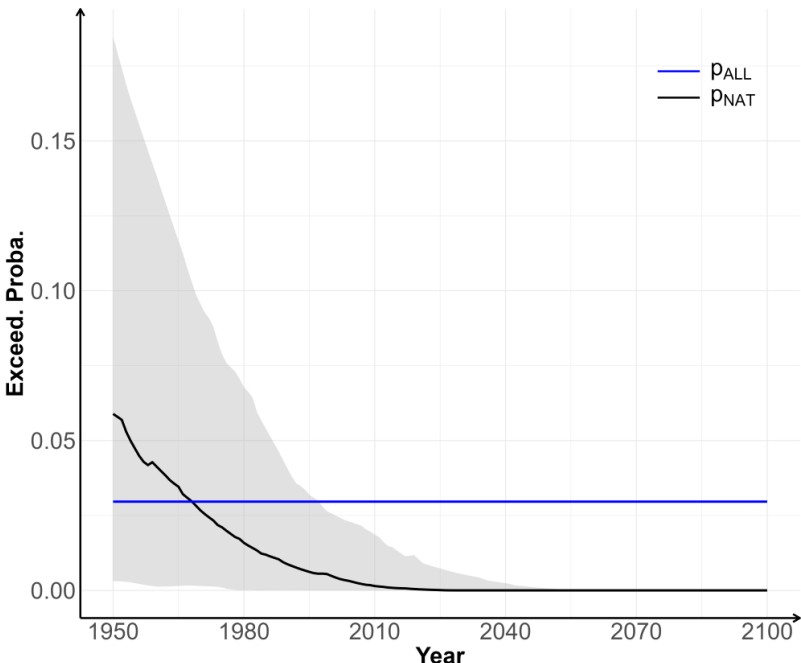

**Figure 6.** Exceedance probability of the FWI-like Landiras-1 wildfire (local and fire-duration set-up) observed (blue) and estimated (black, median $p_{\mathrm{NAT}}$) from NorESM2-LM (r1i1p1f1) simulations under the NAT-only scenario. Note that $p_{\mathrm{ALL}} = p_{\mathrm{OBS}}$ over the full period. Shaded envelope indicates the 80 % parametric-bootstrap confidence interval.

Finally, we examined how ACC made those FWI conditions more or less probable. We first illustrated, for a given model and spatiotemporal scale (Fig. 6), how the exceedance probability $p_{\mathrm{NAT}}$ varies with time relative to $p_{\mathrm{OBS}}$ (and thus $p_{\mathrm{ALL}}$, as the ALL scenario is assumed to mimic observations influenced by all forcings). In this spatiotemporal setup, $p_{\mathrm{NAT}}$ slightly exceeds $p_{\mathrm{ALL}}$ until 1970s although $p_{\mathrm{ALL}}$ remains within the bootstrapping confidence interval. After 1970s, $p_{\mathrm{NAT}}$ becomes systematically lower than $p_{\mathrm{ALL}}$ and tends toward zero after 2040s. We then computed the RR for the Landiras-1 wildfire (Fig. 7) using all models. The RR exhibits a consistent increase from the late 20th century for each model and across all temporal and spatial scales. The multi-model ensembles exceed the reference line $\mathrm{RR} = 1$ (indicating that $p_{\mathrm{NAT}} = p_{\mathrm{ALL}}$) around 1970–1980, with earlier emergences for finer resolutions (Fig. 7d). In 2022, three of the four models yield $\mathrm{RR} > 1$: IPSL-CM6A-LR $\approx 1.5$–3, CanESM5 $\approx 6$–20, and NorESM2-LM $\approx 2$–130; by contrast MIROC6 presents a $RR < 1$ ($\approx 0.4$–0.6). The multi-model ensemble RR lies between $\approx 2$ and 10 in 2022 across the four scales. This ratio reaches the 10–100 range in the latter half of the century, in agreement with the growing influence of ACC. Note that confidence intervals vary strongly across models and scales, reflecting a strong sensitivity to internal variability and parameter uncertainty. Also, RR scores may sometimes exceed 10,000 as from the mid-to-late 21st century due to very low $p_{\mathrm{NAT}}$ approaching zero as shown in Fig. 6.



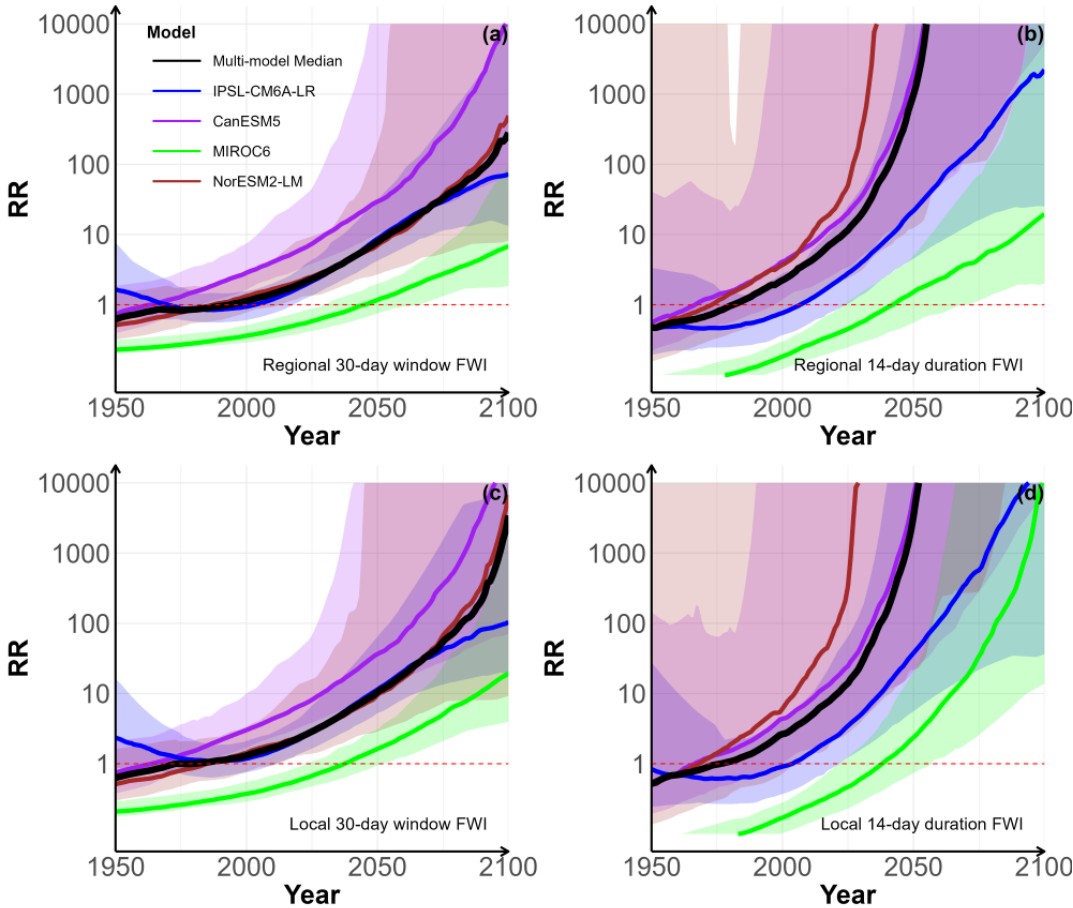

**Figure 7.** Risk ratio (RR) of FWI conditions associated with Landiras-1 wildfire from four CMIP6 models (IPSL-CM6A-LR, CanESM5, MIROC6, NorESM2-LM) and the multi-model median (black) across different scales: (a) regional over 30-day window; (b) regional over 14-day event duration; (c) local over 30-day window; (d) local over 14-day event duration. Shaded envelopes denote 90 % parametric-bootstrap confidence intervals for individual models. All panels use a logarithmic $y$-axis. The red dashed line indicates $RR = 1$ (no anthropogenic influence).

Using the finest spatiotemporal set-up (local over fire duration), we found that ACC contributed approximately to 78 %, 73 %, and 79 % to the FWI conditions associated with Landiras-1, Landiras-2 and La Teste-de-Buch wildfires respectively, and will approach 100 % by mid-21st century (Fig. 8). Note that the signals for Landiras-1 and La Teste-de-Buch are very similar, as the two events occurred approximately over the same period.



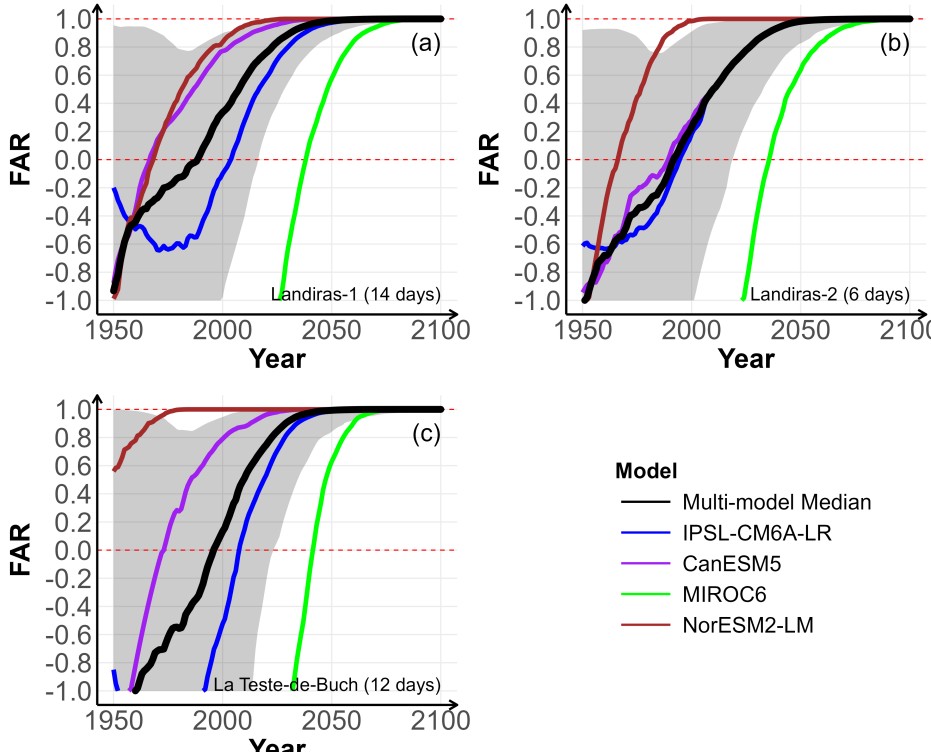

**Figure 8.** Fraction of attributable risk (FAR, $1 - 1/\mathrm{RR}$) using the local FWI over the event-duration window for (a) Landiras-1 (14-day), (b) Landiras-2 (6-day), and (c) La Teste-de-Buch (12-day). The black curve shows the multi-model median across models. The $y$-axis is truncated to $[-1, 1]$; red dashed lines indicate $\mathrm{FAR} = 0$ (no anthropogenic contribution) and $\mathrm{FAR} = 1$ (fully attributable). The shaded envelope indicates the 90 % parametric-bootstrap confidence interval for the multi-model median, estimated by using for each bootstrap replicate $b = 1, \ldots, B$, the multi-model median $\widetilde{\mathrm{FAR}}_b = \mathrm{median}_m\left(\mathrm{FAR}_b^{(m)}\right)$ from the model-specific bootstrap replicates, and by taking the $(0.05, 0.95)$ quantiles of $\{\widetilde{\mathrm{FAR}}_b\}_{b=1}^{B}$.

## 4 Discussion

The spatiotemporal variability of the observed warm-season (May–September) FWI during 1959–2023 has been synthesized into two leading modes. The first mode ($\sim 62$ % of variance explained) shows strong interannual variability due to an alter-
nation between warmer/drier and cooler/wetter years throughout France. Indeed, we found that PC1 was strongly correlated with both temperature and rainfall anomalies over a large portion of western Europe (see Fig. S4 in the Supplement). This mode also features an increased frequency of years with positive FWI anomalies over time. This evolution is consistent with the long-term trend in temperature (Ribes et al., 2022) and drought (Bevacqua et al., 2024) in France, collectively contributing to increased fire weather conditions as observed more broadly in the Mediterranean region (Ruffault et al., 2020; Hetzer et al.,
2024), Europe (Giannaros et al., 2021), and globally (Jain et al., 2022). The second mode (accounting for $\sim 13$ % of variance)





reveals a north–south dipole whose PC correlates with a larger continental-scale dipole in rainfall anomalies, as well as with temperature anomalies south of $45°\,\mathrm{N}$ (see Fig. S5 in the Supplement). This north–south contrast in climate anomalies probably relates to the summer North Atlantic Oscillation, which is, during negative phases, generally associated with blocking events producing cooler/wetter conditions in northern Europe and warmer/drier conditions in the south (Bladé et al., 2012;
Wang et al., 2011; Liu et al., 2025). An underlying long-term trend with more frequent warmer/drier years in the south in recent years was also evident, in agreement with precipitation decreases in southern Europe and slightly wetter conditions in the north as documented in observations (Ruffault et al., 2018; Bevacqua et al., 2024; Tradowsky et al., 2023) as well as in future simulations (Giannaros et al., 2021). Interestingly, both PC1 and PC2 scores indicate that the year 2022 represented a combination of these two leading modes, with unprecedented FWI anomalies over a large portion of France and, orthogonal to
that mode, a latitudinal dipole with higher (lower) FWI in the southern (northern) half of the country.

The unprecedented levels of FWI in 2022 in southwestern France, whether sampled locally or regionally, were conducive to a series of wildfires whose sizes were proportional to the amplitude of FWI anomalies. We found that FWI levels reached their highest amplitude on the day or week of the wildfires, boosted by either synoptic-scale heat waves or local wind bursts, as shown in previous studies over southern Europe (Ruffault et al., 2020) and France (Barbero et al., 2020; Pimont et al., 2021).
We estimated that the conditions observed locally during the three largest wildfires were expected, on average, to occur once every 34, 38, and 101 years for the Landiras-1, Landiras-2, and La Teste-de-Buch wildfires, respectively. The last estimate, relying on extrapolation beyond the observational record, naturally involves substantial uncertainty.

Across spatial scales, Landiras-1 and La Teste-de-Buch wildfires exhibited higher RPs at the local scale than at the regional scale, reflecting the well-known effect of spatial averaging on dampening extremes (Leach et al., 2020; Angélil et al., 2018;
Kirchmeier-Young et al., 2019b). We note that the region is flat, suggesting that those spatial differences cannot be attributed to terrain-related factors. However, Landiras-2 displayed the opposite pattern, with regional RPs exceeding local RPs. This inversion may be due to the fact that Landiras-2 was a rekindling of the Landiras-1 wildfire and was driven mainly by smoldering peat rather than by fire weather conditions. Also, we noticed that some precipitation occurred locally on August 13–14, depressing the local FWI during the 9–14 August interval. Furthermore, even though the regional-scale FWI had a longer return
period, the FWI associated with Landiras-2 was still higher in absolute terms at the local scale. Across temporal scales, all three wildfires exhibited higher RPs during the fire-duration window than during the monthly window, indicating that short-duration, but more acute FWI driven by daily or synoptic meteorological variations also contributed strongly to these wildfires. Note that the wide confidence intervals around the RPs illustrate the uncertainty inherent in GEV fits with limited sample sizes.

Our results indicated that anthropogenic warming has, since the early twenty-first century, markedly increased the probability of occurrence of FWI conditions associated with those wildfires. This is consistent with a growing body of fire-attribution research at the global scale (Jones et al., 2022; Abatzoglou et al., 2025) and at regional scales across parts of the U.S. (Queen et al., 2025; Brown et al., 2023; Williams et al., 2019), Canada (Kirchmeier-Young et al., 2019a, 2024), Australia (van Oldenborgh et al., 2021), the Arctic (Descals et al., 2022), Portugal (Senande-Rivera et al., 2025), and France (Barbero et al., 2020; Lanet et al., 2024). This increase is robust across temporal and spatial scales, and across models. Likewise, the timing
at which the RR exceeds 1 and remains above that threshold seems consistent with the emergence of anthropogenic signals in




simulated fire weather indices since the late twentieth century in southern Europe (Abatzoglou et al., 2019). Our study suggests that climate change increased the risk of such conditions by 2–10 times in 2022 and will continue to do so by several orders of magnitude by the end of the twenty-first century under a medium-level radiative forcing scenario. This validates, across different metrics, fire weather indices, and spatiotemporal scales, the RR obtained by Lanet et al. (2024) using a soil-moisture index as a proxy for fire weather. Interestingly, despite the large spread in RPs across scales, the attribution scores were not found to change substantially across regional and local scales, suggesting that the rarity of FWI conditions does not necessarily relate to the magnitude of the anthropogenic forcing (Cattiaux and Ribes, 2018). Although inter-model differences were evident in terms of amplitude and timing (i.e., the date at which the RR emerges above 1), due to model sensitivity to greenhouse gas emissions, all models point to a substantial increase in 2022-like conditions in future decades, consistent with previous projections of FWI (Fargeon et al., 2020) or FWI-derived fire activity (Pimont et al., 2023b) in France. Also, MIROC6 was found to cross the $\mathrm{RR} = 1$ line later than other models, supporting the findings of Lanet et al. (2024) who detected a lower climate change signal in MIROC6. This is consistent with the lower climate sensitivity of MIROC6 (Tatebe et al., 2019; Forster et al., 2021) due to the radiative forcing of aerosols (Smith et al., 2020) and cloud feedback (Hirota et al., 2022). Finally, our study shows that approximately 70 % of the risk that fire weather conditions reach the levels observed during those wildfires can be attributed to ACC (slightly more than the 49 % found in Lanet et al. (2024) over broader spatial and temporal scales, and the nearly half contribution reported by Barbero et al. (2020) at a multi-decadal scale for Mediterranean France) and that this estimate will reach 100 % by the 2050s. These conclusions apply to other large wildfires (Landiras-2 and La Teste-de-Buch) as well.

We note that the methodology developed here has some limitations. First, our analysis was based solely on meteorological forcing and therefore lacks information on fuels (e.g., forest cover, fuel breaks, and horizontal/vertical continuity). The use of a statistical model that accounts for some of these features (e.g., Firelihood (Pimont et al., 2021)) would produce more realistic estimates. Previous studies have also shown that some wildfires are driven primarily by wind, whereas others are driven by the dryness of climate conditions (Ruffault et al., 2020). Further efforts are thus needed to resolve the respective contributions of fuel moisture and wind forcing, based, for instance, on sub-indices of the FWI (e.g., FFMC, DMC, DC) and complementary atmospheric drivers such as the vapor pressure deficit (VPD) and wind speed, as recently implemented in a probabilistic framework (Castel-Clavera et al., 2025). Our event-attribution analysis was based on three large wildfires in southwestern France. Extending the analysis to the French Mediterranean, which recently experienced the second largest wildfire in France since 1949 (early August 2025; approximately 17,000 ha BA), would be of interest. Although the spread of this wildfire was facilitated by hot and dry conditions alongside strong winds, wildfires in that region are generally associated with higher FWI levels that are expected, on average, once every year or so. Regarding climate simulations, we note that only a single ensemble member (r1i1p1f1) was used for each model. Additional members would reduce internal variability. Further work is also needed to improve the multi-model averaging by introducing weights based on each model's skill over the historical period (Gallo et al., 2025). Finally, we used the SSP2-4.5 scenario in CMIP6, corresponding to a medium level of radiative forcing. Using higher-forcing scenarios (e.g., SSP5-8.5) would yield much higher RR in the future.



The three wildfires examined here burned over multiple days, each undergoing several complete nocturnal cycles. Overnight burning is relatively new in France, where the vast majority of wildfires were historically extinguished within a single day. Further research is thus needed to elucidate (i) the role of nighttime conditions in overnight burning and the extent to which the so-called nighttime barrier is likely to weaken under ACC (Balch et al., 2022), and (ii) the relative contributions of nighttime aridity versus seasonal drought (Luo et al., 2024), as well as the potential role of other meteorological variables such as wind
speed (Chiodi et al., 2025). Such analyses may complement the information provided by traditional daytime-based indicators (e.g., FWI) used in climate–fire or attribution studies.

## 5 Conclusions

This study aimed at quantifying how ACC has already, and will further, alter the probability of fire weather conditions associated with the three largest wildfires of 2022 across France (Landiras-1, Landiras-2, La Teste-de-Buch). First, we found
that warm-season (May–September) FWI has gradually intensified over 1959–2023, especially in Southern France, making the landscape more and more flammable. Second, we found that return periods of FWI associated with extreme wildfires observed in 2022 were scale-dependent and may range for instance from about 5 (at broad scales) to more than 101 years (at finer scales) for La Teste-de-Buch wildfire. Finally, attribution metrics indicated that the FWI levels observed during those wildfires exceed what would be expected in a natural climate and that climate change made those conditions 2 to 10 times more likely in 2022.
Under a moderate-emissions pathway, those FWI conditions will be at least 10–100 times more probable by the end of the century.

*Code availability.* The code used in this study is available from the first author upon reasonable request.

*Data availability.* Wildfire records were obtained from the *Base de Données sur les Incendies de Forêts en France* (BDIFF) (IGN and MASA, 2025) (accessed in 2024). Daily meteorological variables from the SAFRAN atmospheric reanalysis over France (Vidal et al.,
2010) were retrieved from the Météo-France open-data services (https://meteo.data.gouv.fr; accessed in 2024). ERA5 reanalysis fields were downloaded from the Copernicus Climate Data Store (CDS). For reproducibility, we cite the CDS dataset used in this study: *ERA5 hourly data on single levels from 1940 to present* (Copernicus Climate Change Service (C3S), 2018) (accessed in 2024). CMIP6 simulations (historical, DAMIP, and ScenarioMIP) were obtained from the Earth System Grid Federation (ESGF) archive (https://esgf-node.llnl.gov/search/cmip6/; accessed in 2024), following the CMIP6 and MIP design descriptions (Eyring et al., 2016; Gillett et al., 2016; O'Neill et al., 2016).

*Author contributions.* S.Z. performed all analyses and wrote the first draft of the manuscript. S.Z., R.B., F.P., and B.R. jointly contributed to the study design, the interpretation of the results, and the writing and revision of the manuscript.



*Competing interests.* The authors declare that they have no conflict of interest.

*Disclaimer.* The views expressed in this paper are those of the authors and do not necessarily reflect those of their institutions.

*Acknowledgements.* We gratefully acknowledge the support of the Provence–Alpes–Côte d'Azur (PACA) region and the Société du Canal
de Provence (SCP), which funded this research.





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
