# Peer review of "Quantifying the current and future likelihood of the 2022 extreme wildfires weather conditions in France with anthropogenic climate change"

_EGUsphere, 2026_

## Referee Comment (RC1)

Review of Zhu et al. "Quantifying the current and future likelihood of the 2022 extreme wildfires weather conditions in France with anthropogenic climate change"

**Summary**

Zhu et al. present an analysis of the climatic conditions surrounding the extreme 2022 wildfire season in the Southwest of France. The authors use a country-level fire dataset, two different climate re-analysis derived fire weather index (FWI) datasets and CMIP6 model output to quantify how exceptional the 2022 climatic conditions were and how anthropogenic climate change (ACC) has made (and will make) these conditions more likely. The authors find that with increasing spatial-temporal resolution of the FWI, the signal becomes stronger and return period lengths increase by decades, suggesting that aggregating FWI over longer time periods (compared to the time of burning) and larger spatial scales will average out the FWI signal. Furthermore, the climate attribution analysis indicates that already ACC made the FWI conditions during the 2022 fires more than twice as likely compared to the "only natural forcings" run and that by the end of the century these conditions will be 10 to 100 times more likely to occur.

The study presents a nice concise analysis well within the scope of NHESS that is relevant for local stakeholders in the France as well as those in the wider European context. I do not have any concerns with the methods, assumptions or the interpretation of the results. The methodology is not entirely novel as studies of FWI on large spatial scales are common and attribution studies on extreme fire years have also been published before. However, I still think the study presents a scientific advance on the local and regional scale. I have a few major and minor comments which are listed below.

**General comments**

One of the general comments is that there is almost no mention of ignition sources in the manuscript (except a little in the discussion about the ignition of the Landiras-2 fire from peat smoldering). It is not technically the scope of the paper to focus on ignitions because it focusses on the fire weather conditions surrounding these fires, if there are no ignitions sources there will be no fire. It would increase the novelty of the paper dramatically if these were taken into account in the analysis, even if it was a very simple approximation. For example, by making "ignition probability" a function of FWI.

A second major comment concerns the readability of the paper. Generally, the manuscript is written in a very technical style that does not help with reaching a wider readership. If a single sentence in the results contains three abbreviations "FWI", "PC1" and "EOF" that were explained somewhere in the methods, you will lose possibly interested readers. The Conclusions section (5) was surprisingly accessible and I would urge the authors to use the writing style of this section and apply it to the entire manuscript, especially the Abstract and Results sections. It would also help if concepts like PC1 and EOF are shortly explained somewhere in the results, just to inform the reader why these metrics are important and what we can learn from them. The manuscript would benefit from a thorough language revision. I have provided several suggestions in the specific comments, although these are not exhaustive.

**Specific comments**

Title: "extreme wildfires weather", either pick "extreme wildfires in France" or "extreme wildfire weather conditions in France". Both apply to the study.

L7-10 'Our results show that the extremeness of FWI conditions generally increases with the spatiotemporal resolution, with the associated return periods increasing from 6 to 34 years, from 22 to 89 years, and from 6 to 101 years when moving from the coarsest to the finest spatiotemporal scale for the Landiras-1, Landiras-2, and 10 La Teste-de-Buch wildfires, respectively. This sentence is not clear. How does resolution relate to extremeness?

L17 "California in 2025", far more area burned in 2020 in California compared to 2025 (>8 times more, see https://www.fire.ca.gov/incidents/2025 and https://www.fire.ca.gov/incidents/2020) but more infrastructure was damaged in 2025 compared to 2020. I would at least mention 2020 in this sentence.

L22 "[…] and more than 14 times larger than the average in SW France (Fig. 1b)." Please revise this sentence. Separate the stats for France and SW France or rewrite in a different way to make the distinction clearer.

L23 "a small number of wildfires" suggestion: "a small number of large wildfires"

L27 "due to frequent wind shifts causing spread in multiple directions." Might need a reference, local source?

L29-30 "largest wildfire in France since the 1940s." Please cite a reference here, what is the source?

L37 "contributed to reduce fuel moisture content" to a reduced fuel moisture content?

L94 "To quantify how unusual were those conditions" please rephrase

L168-169 "Figure 3 illustrates the first two modes of May–September FWI over the observational 1959–2023 period with the spatial distributions of EOF loadings (left panel) and their corresponding principal components (right panel)." This is a very technical description of the first results that might be improved by bringing the results a bit more descriptive, what is the EOF loading saying and why is it relevant? And why are their PC important? Please rephrase this first section of the results to make it more accessible.

L184-185 "(note that 100% indicates FWI was twice larger than what we would expect from the average local conditions)" please rephrase, FWI is higher not larger. For example: "FWI is twice as high as expected under average local conditions."

L186 "FWI anomalies were that time stronger during" stronger anomalies? Higher or lower please but not stronger.

L190 "indicates that annual maxima of the MA FWI" MA FWI is already maximum annual right? Please rephrase.

L207 "We then computed the RR" please write in full again first time in the results: risk ratio (RR)

L225-226 "Indeed, we found that PC1 was strongly correlated with both temperature and rainfall anomalies over a large portion of western Europe (see Fig. S4 in the Supplement)." Interesting but this is a new result in the Discussion section, please move to results.

L232 "45∘N (see Fig. S5 in the Supplement)." Again a new result in the Discussion, these need to be introduced in the Results.

L241 "The unprecedented levels of FWI in 2022" please rephrase to something like: "The exceptionally high FWI values observed […]" in the original it is not clear if it is high or low.

L243 "their highest amplitude on the day or week of the wildfires" please rephrase, the day or week of ignition? As stated before, some of these larger fires burned for multiple weeks so it is important to be clear here.

**Figures**

Figure 1 Please write burned area (BA) in full in the legend title and y-axis label in panel b. Please remove the coordinates of the bounding box in panel a, they partly overlap with the fire locations and some are unreadable.

Figure 3 many abbreviations in the figure make it very hard to interpret, EOF FWI PC#1, it makes it unnecessary technical. Please reconsider changing the text in this figure that a wide range of audiences immediately can grasp what the figure is showing. Also it is totally unclear to me what a high EOF or low EOF means.

Figure 4 Please write FWI in full on y axis (fire weather index)